# Efficacy of Methylprednisolone Compared to Other Drugs for Pain, Swelling, and Trismus Control after Third Molar Surgery: A Meta-Analysis

**DOI:** 10.3390/healthcare10061028

**Published:** 2022-06-01

**Authors:** Mariana González-Morelos, Lorenzo Franco-de la Torre, Diana Laura Franco-González, Eduardo Gómez-Sánchez, Ángel Josabad Alonso-Castro, Nelly Molina-Frechero, Luis Miguel Anaya-Esparza, Mario Alberto Isiordia-Espinoza

**Affiliations:** 1Hospital Mexico Americano de Guadalajara, Ayuntamiento, Jalisco 44620, Mexico; marianita24gomo@gmail.com; 2Instituto de Investigación en Ciencias Médicas, Departamento de Clínicas, División de Ciencias Biomédicas, Centro Universitario de los Altos, Tepatitlán de Morelos, Jalisco 47620, Mexico; lorfran8888@hotmail.com (L.F.-d.l.T.); diana.franco5288@alumnos.udg.mx (D.L.F.-G.); 3Departamento de Ciencias Fisiológicas, División de Disciplinas Básicas para la Salud, Centro Universitario de Ciencias de la Salud, Universidad de Guadalajara, Jalisco 44340, Mexico; eduardo.gsanchez@academicos.udg.mx; 4Departamento de Farmacia, División de Ciencias Naturales y Exactas, Universidad de Guanajuato, Guanajuato 36000, Mexico; angeljosabad@hotmail.com; 5Laboratorio de Cariología y Medicina Oral, Departamento de Salud, Universidad Autónoma Metropolitana-Xochimilco, Coyoacán, Ciudad de Mexico 04960, Mexico; nmolina@correo.xoc.uam.mx; 6División de Ciencias Agropecuarias y Agrícolas, Centro Universitario de los Altos, Tepatitlán de Morelos, Jalisco 47620, Mexico; luis.aesparza@academicos.udg.mx

**Keywords:** methylprednisolone, non-steroidal anti-inflammatory drugs, postoperative pain, trismus, third molar surgery

## Abstract

The purpose of this systematic review and meta-analysis was to assess the efficacy of methylprednisolone compared to other drugs to control postoperative complications following third molar surgery. PubMed and Google Scholar were used for article searching. Thereafter, the trials meeting the selection criteria and with high methodological quality, according to the Cochrane Collaboration’s risk of bias tool, were included in this study. The inverse variance test and mean difference using the Review Manager Software 5.3 for Windows were used to carry out data analysis. Qualitative analysis shows that methylprednisolone is more effective than NSAIDs, but inferior to dexamethasone, for controlling postoperative complications after third molar removal. The quantitative analysis showed no statistical difference for pain control, while trismus evaluation showed a statistical difference after 7 postoperative days in favor of methylprednisolone, when compared to other drugs. In conclusion, methylprednisolone was more effective for trismus control compared to other drugs after lower third molar surgery.

## 1. Introduction

Several drugs have been used for postoperative complications control following third molar removal [1,2,3,4]. The most used drugs are non-steroidal anti-inflammatory drugs (NSAIDs), opioid analgesics, and corticosteroids [5,6,7,8]. In addition, clinicians—general dentists and specialists—use some strategies for postoperative pain management such as pre-emptive analgesia or multimodal analgesia [9,10,11,12,13]. 

Corticosteroids have several mechanisms of action. For instance, they inhibit the inflammatory process through the inhibition of vasoactive substances (prostaglandins and leukotrienes) and cytokines. These drugs also increase the secretion of lipolytic and proteolytic enzymes [14,15]. Dexamethasone has been the most utilized corticosteroid after third molar surgery [16,17]. Moreover, prednisolone and methylprednisolone have been employed for pain, swelling, and trismus control following third molar extraction. In this sense, a meta-analysis assessed the efficacy of prednisolone and methylprednisolone versus placebo for post-surgical complication control after third molar surgery. However, that meta-analysis did not assess the efficacy of methylprednisolone alone versus other drugs in third molar removal [18]. 

It is important to note that clinical trials have shown inconsistent or conflicting results of methylprednisolone when compared with NSAIDs and corticosteroids in third molar removal [19,20,21,22,23,24,25,26,27,28,29,30,31]. Moreover, some systematic reviews have been conducted without performing a pooled analysis of study data [32]. Considering this, the purpose of this study was to evaluate the efficacy of methylprednisolone alone in comparison with other drugs to control pain, facial inflammation, and mouth opening after third molar surgery.

## 2. Patients and Methods

### 2.1. Design

This study was carried out in the Instituto de Investigación en Ciencias Médicas at the Centro Universitario de los Altos of the University of Guadalajara. The protocol was registered in PROSPERO (Record No. CRD42022314205). In addition, this study followed the PRISMA recommendations for abstracts and declarations for reporting systematic reviews and meta-analyses assessing health care interventions [33,34,35].

### 2.2. Population, Interventions, Control, and Outcome (PICO) Approach 

Inclusion criteria:

Population [36]: Parallel or crossover (split-mouth) clinical trials comparing methylprednisolone and an active control following third molar surgery.

Interventions [36]: Methylprednisolone administration.

Control [36]: Administration of NSAIDs or corticosteroid drugs.

Outcomes [36]: Post-operative pain evaluated with the visual analog scale (VAS), patients needing analgesics, facial inflammation, trismus, and adverse effects.

Exclusion criteria:

Clinical studies reporting a loss of follow-up of more than 20%.

### 2.3. Information Search

The U.S. National Institutes of Health’s National Library of Medicine (PubMed) and Google Scholar were utilized for the identification of clinical trials. The used keywords were “methylprednisolone”, “acetaminophen”, “tenoxicam”, “ketoprofen”, “flurbiprofen”, “diclofenac”, “ibuprofen”, “non-steroidal anti-inflammatory drugs”, “prednisolone”, “dexamethasone”, “corticosteroids”, “third molar surgery”, “post-operative pain”, “facial swelling”, “inflammation”, “trismus”, “mouth opening”, “adverse effects”, and “safety profile”. All investigations evaluating analgesia, swelling, trismus, and/or adverse reactions of methylprednisolone in comparison with an active control published up to 31 December 2021 were eligible. 

### 2.4. Risk of Bias Assessment

The Cochrane Collaboration’s risk of bias tool was used to determine the risk of bias in all studies that met the inclusion criteria [1,2,3,4,37,38,39]. Two independent investigators assessed the risk of bias. The differences between their evaluations were discussed until a consensus was reached [1,2,3,4,37,38,39]. The clinical reports with a high risk of bias were excluded (at least one red ball).

### 2.5. Extraction of Information

The next data were extracted from each clinical investigation: first author and publishing year, design study, treatment groups, sample size (n), dose, pain intensity (VAS; means, SD and, n)—VAS 100 mm were transformed to 10 mm scale—swelling, trismus (means, SD and, n), and adverse effects.

### 2.6. Statistical Analysis

Data were analyzed with the inverse variance statistical method, mean difference, and fixed effects using the Review Manager Software 5.3 for Windows. A global test with a p-value lower than 0.05 of the mean difference and an OR (>1 and within the 95% confidence intervals (95% CI)) were considered statistically significant [1,2,3,4,37,40,41].

## 3. Results

### 3.1. Digital Search

A total of 72 clinical trials were identified in PubMed. No investigations were found in Google Scholar. After eligibility (13 clinical trials), nine clinical studies were included in the qualitative synthesis [19,22,23,24,25,26,29,30,31], and three trials in the meta-analysis [25,29,30] (Figure 1).

### 3.2. Bias Assessment

According to the Cochrane Collaboration’s risk of bias tool, nine clinical trials met the selection criteria [19,22,23,24,25,26,29,30,31]. The main problems of the four excluded studies were the blinding of participants and personnel (performance bias) and blinding of outcome assessment (detection bias) [20,21,27,28] (Figure 2).

### 3.3. Qualitative Evaluation

Seven clinical trials carried out a pre-emptive analgesia approach [19,22,23,24,29,30,31], and two clinical assays used a post-operative analgesia strategy [25,26]. Four clinical investigations used the oral route [19,25,26,29], three trials utilized the IV route [22,23,31], and two studies employed a local route [24,30] for the administration of drugs.

Six clinical trials compared methylprednisolone and NSAIDs following third molar removal [22,23,25,26,29,31]. According to the conclusion of each study, methylprednisolone was better than tenoxicam [22], ketoprofen [23], and diclofenac [25,26] for the management of postoperative pain, facial swelling, and trismus. Furthermore, flurbiprofen was superior to methylprednisolone for pain control [29]. A clinical trial showed no conclusion on methylprednisolone and NSAIDs [31] (Table 1).

Three clinical trials compared the effectiveness of methylprednisolone and dexamethasone after third molar surgery [19,24,30]. Two studies reported results on the control of postoperative pain, swelling, and trismus in favor of dexamethasone [19,30], and one investigation informed similar outcomes for both corticosteroids [24].

### 3.4. Quantitative Assessment

The evaluation of postoperative pain after third molar removal was carried out using data from two clinical trials [25,30] (n = 113). The pooled analysis showed no statistical differences when comparing methylprednisolone and other drugs during the next 6 postoperative days (Figure 3). On the other hand, the mouth opening was assessed with data from two clinical investigations [29,30] (n = 76). The efficacy of methylprednisolone and other drugs was similar from the preoperative evaluation to the second postoperative day. However, the combined assessment of the trismus showed a difference in favor of methylprednisolone, when compared to other drugs after 7 postoperative days (OR = −312; 95% IC=−522 to −1.01; I^2^ = 12%; *p* = 0.004; Figure 4).

## 4. Discussion 

Nine clinical trials were used to carry out, for the first time, a meta-analytical evaluation of the clinical effectiveness of methylprednisolone alone in comparison to NSAIDs and corticosteroids for postoperative pain, facial inflammation, and trismus control after third molar surgery [19,22,23,24,25,26,29,30,31]. This meta-analytical assessment was performed considering high-quality clinical trials only, consistent with the Cochrane Collaboration’s risk of bias tool. According to the results of the qualitative evaluation of our meta-analysis, methylprednisolone was more effective to control pain, swelling, and mouth opening following third molar extraction than NSAIDs [22,23,25,26]. In marked contrast, methylprednisolone exerted a poor control of postoperative pain, facial swelling, and trismus after third molar removal, compared with dexamethasone [19,30].

The quantitative evaluation showed no statistical difference between methylprednisolone and other drugs for pain control after third molar surgery [25,30]. Moreover, the quantitative assessment on trismus found a statistical difference in favor of methylprednisolone when compared to other drugs after 7 postoperative days in third molar surgery [29,30]. Although we found a difference in the evaluation of trismus at 2 and 3 postoperative days, the pooled analysis to compare the efficacy of methylprednisolone and other drugs for trismus control was not possible. Unfortunately, the evaluation of the facial swelling was not carried out because the clinical trials presented different anatomic areas measurement. 

Previously, the clinical efficacy of corticosteroids in oral surgery has been reported. The pooled assessment of a meta-analysis included several clinical trials using methylprednisolone (4/12) to treat postoperative complications following third molar extraction. Such evaluation showed that corticosteroids decrease edema and postoperative pain after oral surgery [42]. Another systematic review and meta-analysis (Markiewicz et al., 2008) show that corticosteroids reduce facial swelling and trismus after third molar surgery [43]. That study included 7/12 clinical trials using methylprednisolone [43]. Falci et al. (2017) carried out a systematic review and meta-analysis to compare the efficacy of dexamethasone and methylprednisolone for controlling pain and trismus after third molar surgery [16]. The authors found that dexamethasone is more effective to control trismus when compared to methylprednisolone, following oral surgery [16]. However, these authors included data from a clinical trial without blinding participants, personnel, and outcome assessment [16]. On the other hand, Nagori et al. (2019) assessed the efficacy of methylprednisolone and prednisolone for control of postoperative complications following third molar surgery [18]. However, the effect of both drugs was included in the same treatment group and compared to placebo in a pooled statistical analysis. Thus, the analgesic, anti-inflammatory, and anti-trismus activity of methylprednisolone alone after third molar surgery was not determined. The main advantage of our study is that the analgesic, anti-inflammatory, and anti-trismus effects of methylprednisolone versus other active controls following third molar extraction were compared through qualitative and quantitative analysis. On the other hand, the main disadvantage of our study was the limited number of studies that presented complete data. That is, means, standard deviation, and sample size, as well as the different ways in which the same clinical indicator was measured—i.e., in the case of adverse effects, some studies report the number of patients with adverse reactions while others the number of adverse effects.

There is evidence of high quality indicating that methylprednisolone is more effective for trismus control after lower third molar surgery than other drugs.

## Figures and Tables

**Figure 1 healthcare-10-01028-f001:**
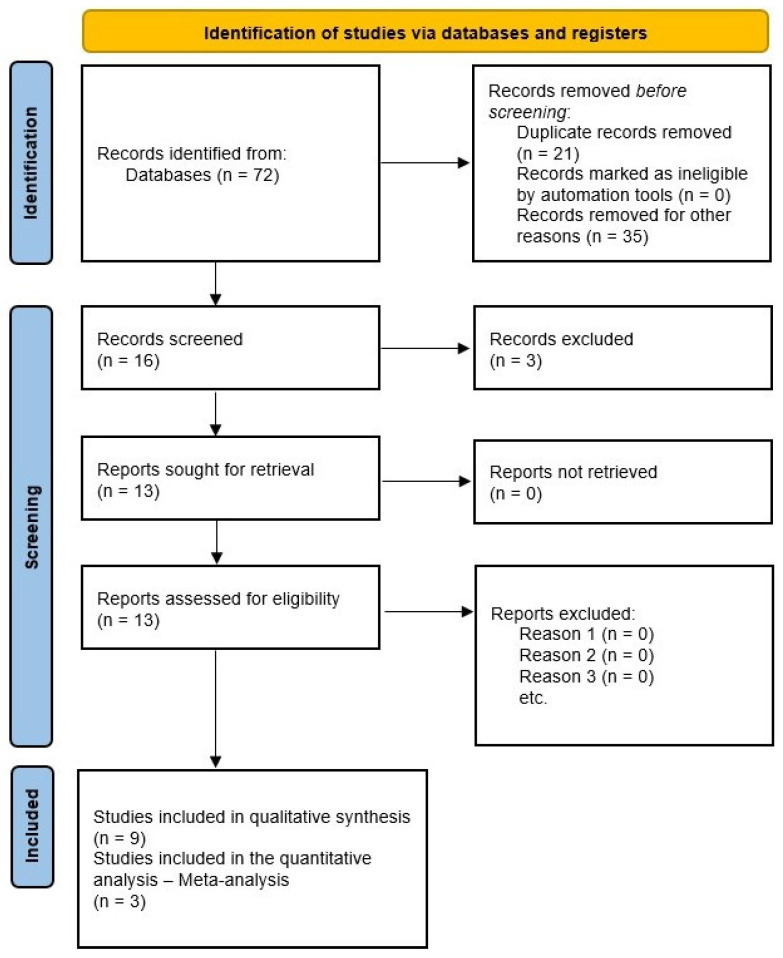
Study flow diagram.

**Figure 2 healthcare-10-01028-f002:**
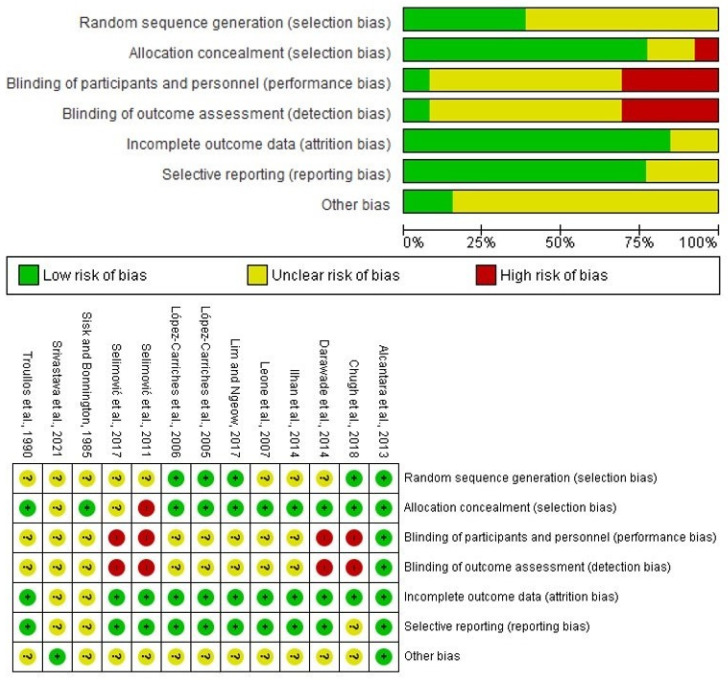
Bias evaluation of the full-text articles [19,20,21,22,23,24,25,26,27,28,29,30,31].

**Figure 3 healthcare-10-01028-f003:**
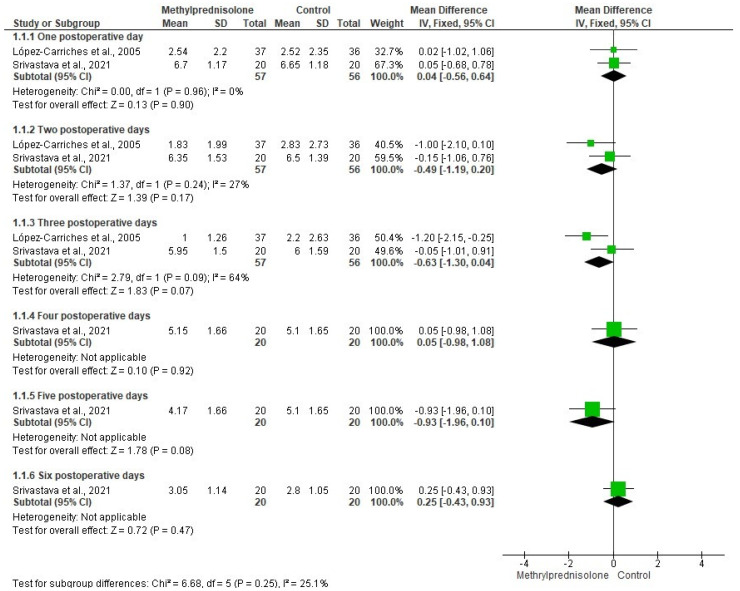
Meta-analysis of the pain intensity using the VAS [25,30].

**Figure 4 healthcare-10-01028-f004:**
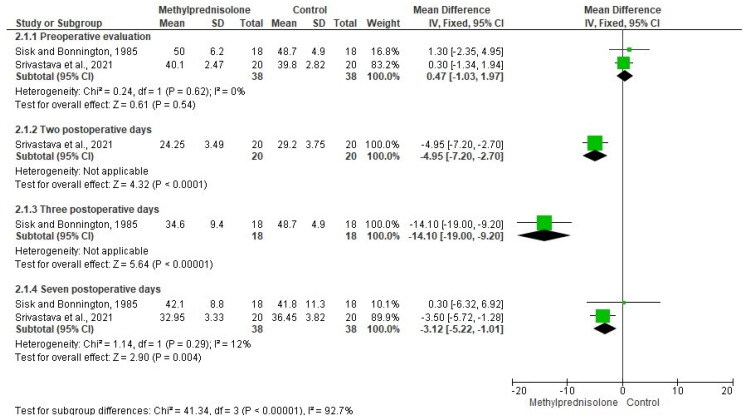
Pooled analysis of the anti-trismus effect (*p* < 0.05) [29,30].

**Table 1 healthcare-10-01028-t001:** Included studies.

ID Study and Study Design	Treatments (n)	Details of Patients, Dental Procedure, and Evaluation	Important Results (Conclusion)
Alcantara et al., 2013 [19]. Randomized, triple-blind, crossover (split-mouth) clinical trial. Preemptive analgesia.	Group A: Methylprednisolone 40 mg (n = 16). Group B: Dexamethasone 8 mg (n = 16). One hour before surgery, patients were given a single oral dose of either drug.	Healthy patients aged 18 and 25 years old, with bilateral mandibular third molars and similar surgical difficulty according to the Pell and Gregory classification (Class II, position B) were included. Patients using drugs with anti-inflammatory and/or analgesic activity within 15 days prior to the study or during the clinical trial were excluded. Extractions were separated by a period of 3 or 4 weeks and carried out by the same surgeon. IANB was performed using Lidocaine 2% and Adrenaline 1:100,000. Acetaminophen 750 mg every 6 h was used as rescue analgesia. Postoperative pain, facial swelling, trismus, and adverse effects were evaluated.	Dexamethasone was better than methylprednisolone regarding facial swelling and mouth opening. However, no differences were observed in the postoperative pain control.
Ilhan et al., 2014 [22]. Randomized, double-blind, parallel, clinical assay. Preemptive analgesia.	Group A: Methylprednisolone 80 mg (n = 20). Group B: Tenoxicam 20 mg (n = 20). Group C: Isotonic sodium chloride (n = 20). All treatments were administered by the IV route.	Patients without any systemic disease and 1 mandibular third molar were included. Patients using drugs with anti-inflammatory and/or analgesic activity within 15 days prior to the study were excluded. All patients were given an IANB using Lidocaine 2% and Adrenaline 1:100,000. Postoperatively, amoxicillin 500 mg, acetaminophen 500 mg (rescue analgesia), and chlorhexidine gluconate (oral antiseptic) were used. Pain, facial edema, mouth opening, and adverse effects were assessed.	Methylprednisolone had a better effect on the mouth opening than tenoxicam. Similar anti-inflammatory and analgesic activity between both drugs was observed.
Leone et al., 2007 [23]. Randomized, double-blind, parallel, clinical study. Preemptive analgesia.	Group A: Methylprednisolone 1mg/kg (n = 46). Group B: Ketoprofen 100 mg (n = 44). All treatments were administered by the IV route.	Patients with multiple third molars were eligible. Subjects with contraindication of anti-inflammatory drugs were ineligible. General anesthesia was performed with propofol and remifentanil. Morphine 0.1 mg/kg was used as postoperative rescue analgesia. Pain, edema, and adverse reactions were measured.	Methylprednisolone was superior to ketoprofen for pain control.
Lim and Ngeow, 2017 [24]. Randomized, double-blind, parallel, clinical investigation. Preemptive analgesia.	Group A: Methylprednisolone 40 mg (n = 20). Group B: Dexamethasone 4 mg (n = 20). Group C: Placebo (n = 20). Drugs were injected into the buccal submucosal area.	ASA-1 patients with mandibular third molars with Class II or position B according to the Pell and Gregory classification were included. Allergic patients or history of adverse effects to the study drugs, acute pericoronitis, chronic use of medications, pregnant or lactating women, and surgical procedure superior to 45 min were excluded. The IANB was performed with Lidocaine 2% and Adrenaline 1:100,000. Postoperative pain, the number of patients needing rescue analgesia, swelling, trismus, and wound healing were evaluated.	A single preoperative dose of methylprednisolone and dexamethasone were equally effective for control of postsurgical sequels following third molar surgery.
López-Carriches et al., 2005 [25]. Randomized, double-blind, parallel, clinical trial. Postoperative analgesia.	Group A: Methylprednisolone 4 mg (n = 37). Group B: Diclofenac 50 mg (n = 36). Drugs were given orally every 8 h.	ASA-1 patients, aged 18 to 42 years old, and no toothache were included. Patients with systemic disease, pregnant or lactating women, with dental symptomatology, and patients taking anti-inflammatory drugs were excluded. Drugs used for the IANB were not reported. Amoxicillin 750 mg and metamizole 575 mg; both drugs every 8 h for 3 days were used. Pain, the number of patients taking rescue analgesic medication, and adverse effects were recorded.	Patients receiving methylprednisolone had a lower pain score in comparison to those who were given diclofenac.
López-Carriches et al., 2006 [26]. Randomized, double-blind, parallel, clinical study. Postoperative analgesia.	Group A: Methylprednisolone 4 mg (n = 37). Group B: Diclofenac 50 mg (n = 36). Drugs were given orally every 8 h.	ASA-1 patients, aged 18 to 42 years old, and no toothache were included. Systemic disease, pregnant or lactating, dental symptomatology, and patients taking anti-inflammatory drugs were excluded. Drugs used for the IANB were not reported. Amoxicillin 750 mg and metamizole 575 mg; both drugs every 8 h for 3 days were used. Facial swelling, mouth opening, and adverse effects were assessed.	Methylprednisolone was better than diclofenac for facial edema and trismus control.
Sisk and Bonnington, 1985 [29]. Randomized, double-blind, parallel, clinical investigation. Preemptive analgesia.	Group A: Methylprednisolone 125 mg (n = 18). Group B: Flurbiprofen 50 mg (n = 18). Group C: Placebo (n = 19). All treatments were administered orally 30 min before surgery.	Patients aged 16-35 years old who needed extraction of four mandibular third molars were included. Patients with uncontrolled systemic disease, patients requiring additional antibiotic therapy, and/or smokers were excluded. Sedation (diazepam ≤ 0.4 mg/kg) and local anesthesia (2% lidocaine and adrenaline 1:100,000) were used. Paracetamol-codeine rescue analgesic medication was used. Postoperative pain, swelling, mouth opening, and adverse effects were measured.	Flurbiprofen had better analgesic activity than methylprednisolone.
Srivastava et al., 2021 [30]. Randomized, triple-blind, crossover (split-mouth), clinical assay. Preemptive analgesia.	Group A: Methylprednisolone 40 mg (n = 20). Group B: Dexamethasone 8 mg (n = 20). Both drugs were given (injection) into the masseter muscle.	ASA-1 patients aged 18 to 35 who required the extraction of both mandibular third molars with similar surgical difficulty were included. Patients needing rescue analgesia within the first 6 postoperative hours were excluded. Study methodology did not report any drug used for carrying out the local anesthesia or any rescue analgesic medication. Postoperative pain, swelling, mouth opening, and adverse effects were measured.	Dexamethasone was better than methylprednisolone for pain control after third molar surgery.
Troullos et al., 1990 [31]. Randomized, triple-blind, parallel, clinical trial. Preemptive and postoperative analgesia.	The first study (Troullos et al., 1990a). Group A: Methylprednisolone 125 mg (n = 20). Group B: Flurbiprofen 100 mg (n = 20). Group C: Placebo (n = 20). All treatments were administered 30 min before surgery using the IV route and were taken every 6 h for 3 days. The second study (Troullos et al., 1990b). Group A: Methylprednisolone 125 mg (n = 15). Group B: Ibuprofen 600 mg (n = 15). Group C: Placebo (n = 15). All treatments were administered 30 min before surgery using the IV route and were taken every 6 h for 3 days.	Patients with the absence of systemic disease, and no history of allergy to the study drugs were included. Patients with pain preoperatively were excluded. Sedation (diazepam) and local anesthesia (lidocaine 2% and adrenaline 1:100,000) were employed. Paracetamol 650 mg and codeine 60 mg were utilized as analgesics. Pain, swelling, and adverse effects were recorded.	The study conclusion informed no comparison of the clinical efficacy of methylprednisolone versus flurbiprofen, or diclofenac.

## Data Availability

Data for this meta-analysis were obtained from the following articles.
Alcântara, C.; Falci, S.; Oliveira-Ferreira, F.; Santos, C.; Pinheiro, M. Pre-emptive effect of dexamethasone and methylprednisolone on pain, swelling, and trismus after third molar surgery: A split-mouth randomized triple-blind clinical trial. *Int. J. Oral Maxillofac. Surg.*
**2014**, *43*, 93–98.Ilhan, O.; Agacayak, K.S.; Gulsun, B.; Koparal, M.; Gunes, N. A comparison of the effects of methylprednisolone and tenoxicam on pain, edema, and trismus after impacted lower third molar extraction. *Med. Sci. Monit.* **2014**, *20*, 147–152.Leone, M.; Richard, O.; Antonini, F.; Rousseau, S.; Chabaane, W.; Guyot, L.; Martin, C. Comparison of methylprednisolone and ketoprofen after multiple third molar extraction: A randomized controlled study. *Oral Surg. Oral Med. Oral Pathol. Oral Radiol. Endodontol.*
**2007**, *103*, e7–e9.Lim, D.; Ngeow, W.C. A Comparative study on the efficacy of submucosal injection of dexamethasone versus methylprednisolone in reducing postoperative sequelae after third molar surgery. *J. Oral Maxillofac. Surg*. **2017**, *75*, 2278–2286.López-Carriches, C.; Martínez-González, J.M.; Donado-Rodríguez, M. Analgesic efficacy of diclofenac versus methylprednisolone in the control of postoperative pain after surgical removal of lower third molars. *Med. Oral Patol. Oral Cir Bucal.*
**2005**, *10*, 432–439.López-Carriches, C.; Martínez-González, J.M.; Donado-Rodríguez, M. The use of methylprednisolone versus diclofenac in the treatment of inflammation and trismus after surgical removal of lower third molars. *Med. Oral Patol. Oral Cir. Bucal.*
**2006**, *11*, E440–E445.Sisk, A.L.; Bonnington, G.J. Evaluation of methylprednisolone and flurbiprofen for inhibition of the postoperative inflammatory response. *Oral Surg. Oral Med. Oral Pathol.* **1985**, *60*, 137–145.Srivastava, N.; Shetty, A.; Kumar, P.; Rishi, D.; Bagga, V.; Kale, S.G. Comparison of Preemptive Effect of Dexamethasone and Methylprednisolone After Third Molar Surgery: A Split-Mouth Randomized Triple-Blind Clinical Trial. *J. Maxillofac. Oral Surg.* **2020**, *20*, 264–270.Troullos, E.S.; Hargreaves, K.M.; Butler, D.P.; Dionne, R.A. Comparison of nonsteroidal anti-inflammatory j., ibuprofen and flurbiprofen, with methylprednisolone and placebo for acute pain, swelling, and trismus. *J. Oral Maxillofac. Surg.* **1990**, *48*, 945–952. Alcântara, C.; Falci, S.; Oliveira-Ferreira, F.; Santos, C.; Pinheiro, M. Pre-emptive effect of dexamethasone and methylprednisolone on pain, swelling, and trismus after third molar surgery: A split-mouth randomized triple-blind clinical trial. *Int. J. Oral Maxillofac. Surg.*
**2014**, *43*, 93–98. Ilhan, O.; Agacayak, K.S.; Gulsun, B.; Koparal, M.; Gunes, N. A comparison of the effects of methylprednisolone and tenoxicam on pain, edema, and trismus after impacted lower third molar extraction. *Med. Sci. Monit.* **2014**, *20*, 147–152. Leone, M.; Richard, O.; Antonini, F.; Rousseau, S.; Chabaane, W.; Guyot, L.; Martin, C. Comparison of methylprednisolone and ketoprofen after multiple third molar extraction: A randomized controlled study. *Oral Surg. Oral Med. Oral Pathol. Oral Radiol. Endodontol.*
**2007**, *103*, e7–e9. Lim, D.; Ngeow, W.C. A Comparative study on the efficacy of submucosal injection of dexamethasone versus methylprednisolone in reducing postoperative sequelae after third molar surgery. *J. Oral Maxillofac. Surg*. **2017**, *75*, 2278–2286. López-Carriches, C.; Martínez-González, J.M.; Donado-Rodríguez, M. Analgesic efficacy of diclofenac versus methylprednisolone in the control of postoperative pain after surgical removal of lower third molars. *Med. Oral Patol. Oral Cir Bucal.*
**2005**, *10*, 432–439. López-Carriches, C.; Martínez-González, J.M.; Donado-Rodríguez, M. The use of methylprednisolone versus diclofenac in the treatment of inflammation and trismus after surgical removal of lower third molars. *Med. Oral Patol. Oral Cir. Bucal.*
**2006**, *11*, E440–E445. Sisk, A.L.; Bonnington, G.J. Evaluation of methylprednisolone and flurbiprofen for inhibition of the postoperative inflammatory response. *Oral Surg. Oral Med. Oral Pathol.* **1985**, *60*, 137–145. Srivastava, N.; Shetty, A.; Kumar, P.; Rishi, D.; Bagga, V.; Kale, S.G. Comparison of Preemptive Effect of Dexamethasone and Methylprednisolone After Third Molar Surgery: A Split-Mouth Randomized Triple-Blind Clinical Trial. *J. Maxillofac. Oral Surg.* **2020**, *20*, 264–270. Troullos, E.S.; Hargreaves, K.M.; Butler, D.P.; Dionne, R.A. Comparison of nonsteroidal anti-inflammatory j., ibuprofen and flurbiprofen, with methylprednisolone and placebo for acute pain, swelling, and trismus. *J. Oral Maxillofac. Surg.* **1990**, *48*, 945–952.

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
