# Peer review of "Efficacy of Methylprednisolone Compared to Other Drugs for Pain, Swelling, and Trismus Control after Third Molar Surgery: A Meta-Analysis"

_healthcare, 2022, doi:10.3390/healthcare10061028_

Round 1
Reviewer 1 Report
Dear Authors, thank you very much for submitting your manuscript. In general, the paper is well written, clear and the topic is well explained. I think that there are any corrections that could improve the work:
- in the materials and methods specify that the systematic review has followed the PRISMA statement criteria;
- In the Table 1, the third column is not clear and it is difficult to read. I suggest to resume in standard sentences this column.
- Explain in a better way the statistical results because the section "Quantitative assessment" is too short and unclear.
Author Response
The attached document contains the reply to the reviewer.

Reviewer 2 Report
Thank you for this helpful contribution. I applaud the effort of promoting studies for the investigation of drugs efficacy. I appreciate their methods including presentation of intervention description and the flow diagram. In addition to the below comments, the manuscript would benefit from additional proof reading for grammar and syntax.
Results. Tables. Information is shown in duplicate. If it appears in the table, delete the information from the text.
Conclusion????? Conclusion must respond to the main aim of the study. It is advisable that conclusion be clear and concise. Therefore, it is not recommended that its length be so long.
Congratulation again!!!
Author Response

(The authors gave the same response as above.)

Reviewer 3 Report
The authors report a systematic review of the use of methylprednisolone after third-molar extraction surgery. The systematic review is well conducted and selection criteria are consistent.
In general, the entire text needs to be proof-read for English language.
The discussions should be expanded somewhat by analyzing the results of the meta-analysis in comparison with the studies in the literature. What do other studies say about methylprednisolone? They confirm or deny your results.
In the discussions the authors report that Falci et al included a study without blinding of the participants, however, from the risk of bias analysis it emerges that 12 out of 13 studies had problems with blinding, 4 had even high risk, could you please comment on this data?
Thanks
Author Response
The attached document contains the reply to the three reviewers.

Round 2
Reviewer 3 Report
I have no additional comments. Congrats